



**Evaluating the Recent "2+26" Regional Strategy for Air Quality**
**Improvement During Two Orange Air Pollution Alerts in Beijing:**
**variations of PM$_{2.5}$ concentrations, source apportionment, and the**
**relative contribution of local emission and regional transport**
**Ziyue Chen[1,2], Danlu Chen[1], Nianliang Cheng[3,4], Yan Zhuang[1], Mei-Po Kwan[5,6], Bin Chen[7],**
**Bo Zhao[8], Lin Yang[9*], Bingbo Gao[10], Ruiyuan Li[1], Bing Xu[11*]**
[1] State Key Laboratory of Earth Surface Processes and Resource Ecology, College of Global and
Earth System Sciences, Beijing Normal University,19 Xinjiekou Street, Haidian, Beijing 100875,
China
[2]Joint Center for Global Change Studies, Beijing 100875, China
[3]Chinese Research Academy of Environmental Sciences, Beijing 100012, China
[4]Beijing Municipal Environmental Monitoring Center, Beijing 100048, China
[5]Department of Geography and Geographic Information Science, University of Illinois at
Urbana-Champaign, Urbana, IL 61801, USA
[6]Department of Human Geography and Spatial Planning, Utrecht University, 3584 CB Utrecht,
The Netherlands
[7]Department of Land, Air and Water Resources, University of California, Davis, CA 95616, USA
[8]College of Earth, Ocean, and Atmospheric Sciences, Oregon State University, Oregon, USA
[9]School of Geographic and Oceanographic Sciences, Nanjing University, Nanjing,210023, China
[10]National Engineering Research Center for Information Technology in Agricultre,11 Shuguang
Hua yuan Middle Road, Beijing 100097, China
[11]Ministry of Education Key Laboratory for Earth System Modeling, Department of Earth System
Science, Tsinghua University, Beijing 100084, China
[*]To whom correspondence should be addressed. Email: yanglin@nju.edu.cn or
bingxu@tsinghua.edu.cn
**Abstract**
To comprehensively evaluate the effects of the recent "2+26" regional strategy for air quality
improvement, we compared the variations in PM$_{2.5}$ concentrations in Beijing during four pollution
episodes with different emission-reduction strategies. The "2+26" strategy implemented in March
2018 led to a mean PM$_{2.5}$ concentrations of 16.43% lower than that during the pollution episode in
March 2013, when no specific emission-reduction measures were in place. The same "2+26"
strategy implemented in November 2017 led to a mean PM$_{2.5}$ concentrations of 32.70% lower than
that during the pollution episode in November 2016, when local emission-reduction measures
were implemented. The results suggested that the effects of the "2+26" regional
emission-reduction measures on PM$_{2.5}$ reductions were influenced by a diversity of factors and
could differ significantly during specific pollution episodes. Furthermore, we found the



proportions of sulfate ions decreased significantly and nitrate ions were the dominant $PM_{2.5}$
components during the two "2+26" orange alert periods. Meanwhile, the relative contributions of
coal combustion to $PM_{2.5}$ concentrations in Beijing during the pollution episodes in March 2013,
November 2016, November 2017 and March 2018 was 40%, 34%, 28% and 11% respectively,
indicating that the recent "Coal to Gas" project and the contingent "2+26" strategy led to a
dramatic decrease in coal combustion in the Beijing-Tianjin-Hebei Region. On the other hand, the
relative contribution of vehicle exhaust during the "2+26" orange alert periods in November 2017
and March 2018 reached 40% and 54% respectively. The relative contribution of local emission to
$PM_{2.5}$ concentrations in Beijing also varied significantly and ranged from 49.46% to 89.35%
during the four pollution episodes. These results suggested that the "2+26" regional
emission-reduction strategy should be implemented with red air pollution alerts during heavy
pollution episodes to intendedly reduce the dominant contribution of vehicle exhausts to $PM_{2.5}$
concentrations in Beijing, while specific emission-reduction measures should be implemented
accordingly for different cities within the "2+26" framework.
**Keywords**: **Air pollution alert; Regional integration; Emission reduction;**
**WRF-CAMx; Beijing; "2+26".**

## 1 Introduction

In January 2013, a severe haze episode with the highest concentration of hourly fine particulate
matter with a diameter of less than 2.5 micrometers ($PM_{2.5}$) occurred in Beijing (886 $\mu g/ m^3$),
which attracted worldwide attention. Since 2013, Beijing, located in the Beijing-Tianjin-Hebei
region, has been a heavily polluted area in China that suffers from continuous haze episodes
associated with high concentrations of $PM_{2.5}$, especially in winter. Given the significant negative
influence of $PM_{2.5}$ on public health (Garrett and Casimiro, 2011; Guaita et al., 2011; Pasca et al.,
2014; Li et al., 2015), the air quality management authority in Beijing has put growing emphasis
on long-term environmental protection policies, including shutting down polluting factories and
limiting vehicle use through license plate rules. However, total emissions of airborne pollutants
remain at very high levels in Beijing, leading to frequent heavy pollution episodes (Guo et al.,
2012). To mitigate this problem, contingent emission-reduction measures, in addition to regular
environmental policies, are necessary in Beijing in order to improve local air quality during air
pollution episodes.
In 2013, the Beijing Municipal Government published the "Heavy Air Pollution Contingency
Plan" and revised this plan in 2015 to better manage air quality during pollution episodes.
According to the predicted concentrations of different airborne pollutants and the duration of
pollution episodes, there are four levels of air pollution alerts for Beijing, which are blue, yellow,
orange, and red alerts. Specific emission-reduction measures are implemented when each type of
air pollution alerts is in effect. The red alert is the most stringent level of air pollution alerts and
predicts severe air pollution episodes (Air Quality Index [AQI] >300) that will last for more than
three days. Emission-reduction measures during red alerts mainly include the implementation of
the odd-even license plate policy (only about half of all of the cars in Beijing is allowed to run



within the fifth-ring district in each day), the suspension of all outdoor construction work and
temporary shutdown of listed polluting factories. The orange alert predicts heavy air pollution
episodes (AQI >200) that will last for more than three days. Emission-reduction measures during
orange alerts mainly include forbidding vehicles that cannot meet the Environmental Standard
Levels I and II, the suspension of specific outdoor work (e.g., painting) and temporary shutdown
of listed polluting factories (the list for red alerts includes more factories than that for orange
alerts). The blue and yellow alerts predict heavy air pollution episodes that will last for more than
one and two days respectively. There are very few compulsory emission-reduction measures for
blue and yellow alerts and most emission-reduction measures are suggestive. The characteristics
and effects of these emission-reduction measures during alert periods have been massively studied
(Zhong, J. et al., 2017; Zhang, Z. et al.,2017; Wang, X. et al., 2017; Zeng, W. et al., 2018; Shang,
X. et al., 2018). However, previous emission-reduction measures during orange and red alerts
were solely conducted in a specific city (e.g., Beijing) while regional emission-reduction measures
implemented simultaneously in many adjacent cities have rarely been implemented and evaluated.
Although the peak $PM_{2.5}$ concentrations in Beijing could be reduced by 20% through strict
emission-reduction measures (Cheng et al., 2017), local $PM_{2.5}$ concentrations remained at very
high levels during red alert periods. This is mainly attributed to the regional transport of airborne
pollutants from neighboring cities to Beijing (Chen et al., 2016). Therefore, regional integration
has become one of the major solutions for further reducing $PM_{2.5}$ concentrations in Beijing during
heavy pollution episodes. To promote this strategy, the Ministry of Environmental Protection of
the People's Republic of China released the "2017 Air Pollution Prevention and Management Plan
for the Beijing-Tianjin-Hebei Region and its Surrounding Areas" (MEP, 2017). This plan suggests
that Beijing, Tianjin, eight cities in Hebei Province, four cities in Shanxi Province, seven cities in
Shandong Province and seven cities in Henan Province (2+26) constitute the regional network
involved in the long-distance transport of airborne pollutants surrounding Beijing. Therefore,
during heavy pollution episodes, unified emission-reduction measures should be carried out in
these cities simultaneously to reduce extremely high $PM_{2.5}$ concentrations in Beijing.
Since the launch of the "2+26" plan, Beijing experienced two pollution episodes in November
2017 and March 2018, when MEP released two orange alerts and implemented corresponding
emission-reduction measures in all 28 cities simultaneously. The two orange alerts were the first
two attempts of the "2+26" plan to reduce $PM_{2.5}$ concentrations in Beijing. To better evaluate this
"2+26" regional strategy and for a comprehensive comparison, we also included in this study two
other pollution episodes in Beijing: November 2016 (with local emission-reduction measures) and
March 2013 (with no emission-reduction measure). We first analyzed the variations in $PM_{2.5}$
concentrations in Beijing during the four pollution episodes. Following this, we quantified the
component and sources of the $PM_{2.5}$. Based on source apportionment, we further quantified the
relative contributions of local emissions and regional transport to $PM_{2.5}$ concentrations in Beijing
during these four pollution episodes. The methodology and findings of this research not only holds
practical significance for further improving the "2+26" regional strategy, but also shed some light
on the regional integration of air quality management in other parts of China.



## 2 Materials and methods

### 2.1 Study sites

Beijing is located at the northwestern edge of the North China Plain. It is surrounded by mountains on three sides, resulting in a geographical condition unfavorable for the dispersion of airborne pollutants. Therefore, air pollution episodes have been frequently witnessed in Beijing since 2013, especially in winter. Based on large-scale field-experiments and model simulation, MEP (2017) pointed out that 28 cities formed a regional transport network of airborne pollutants, which influenced local PM$_{2.5}$ concentrations in Beijing significantly. These 28 cities include two municipalities directly under the central government, Beijing and Tianjin and another 26 neighboring cities surrounding Beijing, which are Shijiazhuang, Tangshan, Lang fang, Baoding, Cangzhou, Hengshui, Xingtai and Handan in Hebei Province, Taiyuan, Yangquan, Changzhi and Jincheng in Shanxi Province, Jinan, Zibo, Jining, Dezhou, Liaocheng, Binzhou and Heze in Shandong Province, Zhengzhou, Kaifeng, Anyang, Hebi, Xinxiang, Jiaozuo and Puyang in Henan Province. The locations of these cities are shown in Fig 1. These 26 cities, especially those cities located in the Hebei provinces, are mainly industrial cities that consume a large amount of coals and produce massive amounts of airborne pollutants. To comprehensively understand the effects of the "2+26" regional strategy for air quality improvement in Beijing, all these 28 cities were selected as study sites for this research.

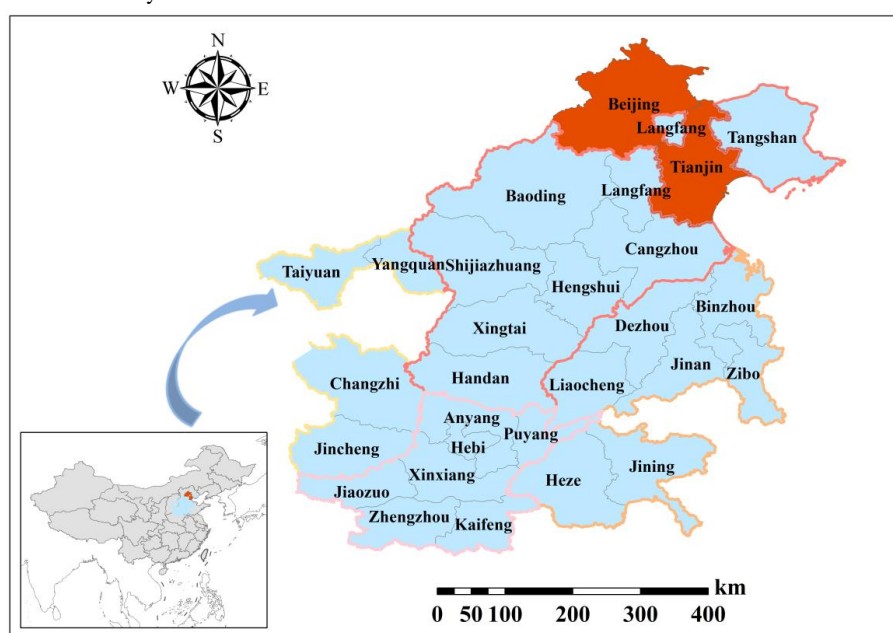

**Fig 1. Geographical locations of the 28 cities within the "2+26" regional integration framework**



### 2.2 Data Sources

### 2.2.1 Ground PM$_{2.5}$ and meteorological observation data

The data of major airborne pollutants for this research were collected from the website PM25.in. This website assembles official data of major airborne pollutants provided by the China National Environmental Monitoring Center (CNEMC) and publishes hourly air quality information for 367 monitored cities in China, which include all of the 28 cities in the "2+26" framework. By using a specific API (Application Programming Interface) provided by PM25.in, we collected hourly pollutant data (e.g., PM$_{2.5}$, CO, NO$_2$, O$_3$) for these 28 cities. The hourly average concentration of each pollutant for one city is calculated by averaging the hourly value measured at all available observation stations within the city. For the following analysis, we employed time series air quality data covering all four pollution periods: from 0 AM, November 24$^{th}$ to 12 PM, November 27$^{th}$, 2016; from 0 AM, November 4$^{th}$, 2017 to 12 PM, November 7$^{th}$, 2017; from 0 AM, March 14$^{th}$ to 12 PM, March 17$^{th}$, 2013; from 0 AM March 11$^{th}$ to 12 PM, March 14$^{th}$, 2018.

In addition to large-scale meteorological data for the following simulation, we also employed ground observation data to compare meteorological conditions during these four pollution episodes. Meteorological data for this research were collected at the Guanxiangtai Station in Beijing and were downloaded from the Department of Atmospheric Science of the University of Wyoming (http://weather.uwyo.edu/upperair/sounding.html). Based on the comparison of the meteorological data, we could ascertain whether large variations in meteorological conditions existed between the four pollution episodes, as a potential influencing factor of the variations in PM$_{2.5}$ concentrations.

### 2.2.2 PM$_{2.5}$ component Data

To comprehensively understand the component of PM$_{2.5}$ during the four pollution episodes, we collected PM$_{2.5}$ sample data at the DongSi Station for further analysis. These PM$_{2.5}$ sample data were collected during the pollution episodes in March, 2013, November 2016, November 2017 and March 2018 respectively. We employed an URG-9000B Ambient Ion monitor (Thermo Fisher Scientific), which includes two Dionex ICS-90 ion chromatography systems (DIONEX, US), to detect water soluble ion Na$^+$, Mg$^{2+}$, Ca$^{2+}$、K$^+$、NH$_4^+$、Cl$^-$、SO$_4^{2-}$、NO$_3^-$、PO$_4^{3-}$. The original temporal resolution for ion detection was 15 minutes and for the comparison with other components, the resolution for water-soluble ion detection was averaged to an hour. The organic carbon concentration of PM$_{2.5}$ was analyzed using the OC/EC organic carbon analyzer (sunset lab model 5l) and the temporal resolution for carbon detection was an hour. The in-depth analysis of PM$_{2.5}$ component provides significant reference for understanding the evolution and sources of PM$_{2.5}$ during the pollution episodes.

### 2.3 Method

### 2.3.1 Simulation Models

We employed the WRF-CAMx model for simulating the effects of emission reduction measures on the reduction of major airborne pollutants. The WRF-CAMx includes three models: the





middle-scale meteorology model (WRF), the source emission model (SMOKE)
(https://www.cmascenter.org/cmaq/) and air quality model (CAMx) (http://www.camx.com/). The
WRF model provided the meteorological field for the analysis. The CAMx model has been widely
used for simulating the evolution of air pollution episodes (An et al., 2007; Liu et al., 2010) and
was employed for simulating the variations of airborne pollutants in this study (ENVIRON, 2013).
In this research, the central point for the CAMx was set at the coordinate (35 °N, 110 °E) and
bi-directional nested technology was employed, producing two layers of grids with a horizontal
resolution of 36 km and 12 km respectively. The first layer of the grids has a 36km resolution and
200*160 cells covering most areas in East Asia (including Japan, South Korea, China, North
Korea, and other countries). The second layer of the grids has a 12km resolution and 120*102
cells, based on the lambert map projection and standard latitude lines 24 °N and 46 °N, covering the
North China Plain, which includes the Beijing-Tianjin-Hebei region, Shandong and Henan
Provinces. The vertical layer was divided into 20 unequal layers, eight of which were of
less-than-1km distance to the ground for better featuring of the structure of atmospheric boundary.
Airborne pollutants in CAMx were simulated according to some physical and chemical
mechanisms, including: a) horizontal advection scheme (PPM), b) implicit Euler vertical
convection scheme, c) the horizontal diffusion of K theory, d) Saprc99 gas-phase chemical
mechanism, and e) EBI calculation method. The initial and boundary conditions for simulating
airborne pollutants were set using the default CAMx profiles. For better simulating the pollution
process with longer time series, the simulation period was set as the entire March 2013, November
2016, November 2017, and March 2018. For the first running of this model, a spin-up period of 5
days was set to simulate the initial field and the following initial field was decided by the output of
previous simulations. Hence, the accumulation effects of emission sources have been
comprehensively considered and the influence of uncertain initial conditions has been reduced
significantly.
We employed ARW-WRF3.2 to simulate the meteorological field. The setting of the center and the
bi-directional nest for the WRF was similar to that of the CAMx as mentioned above. There were
35 vertical layers for the WRF and the outer layer provided boundary conditions of the inner layer.
The meteorological background field and boundary information with a GFS resolution of $1°\times1°$
and temporal resolution of 6h were acquired from NCAR (National Center for Atmospheric
Research, https://ncar.ucar.edu/) and NCEP (National Centers for Environmental Prediction)
respectively. The terrain and underlying surface information was obtained from the USGS 30s
global DEM (https://earthquake.usgs.gov/). The output from the WRF model was interpolated to
the region and grid for the CAMx model using the Meteorology-Chemistry Interface Processor
(MCIP, https://www.cmascenter.org/mcip). The meteorological factors used for this model include
temperature, air pressure, humidity, geopotential height, zonal wind, meridional wind,
precipitation, boundary layer heights and so forth. An estimation model for terrestrial ecosystem
MEGAN (http://ab.inf.uni-tuebingen.de/software/megan/) was employed to process the natural
emissions. Anthropogenic emission data were from the Multi-resolution Emission Inventory for
China, MEIC $0.5°\times0.5°$ emission inventory (http://www.meicmodel.org/) and Beijing emission
inventory (http://www.cee.cn/). We input the processed natural and anthropogenic emission data
into the SMOKE model and acquired comprehensive emission source files.



**Table 1 Sources of Emission inventory**

| Airborne Pollutants | Sources | Data description |
| --- | --- | --- |
| $PM_{2.5}$, BC, OC | MEIC | Resolution: 0.5 °×0.5 ° |
| $SO_2$ | Survey of Emission sources | Point sources, Polygon sources |
| NOx | Survey of Emission sources | Point sources, Polygon sources |
| $PM_{10}$ | Survey of Emission sources | Point sources, Polygon sources |
| $NH_3$ | MEIC | Resolution: 1 °×1 ° |
| Anthropogenic VOCs | MEIC | Resolution: 0.5 °×0.5 ° |
| Natural VOCs | MEGAN | Corresponding Grid data |

**2.3.2 Source Apportionment**
PSAT (Particulate Matter Source Apportionment Technology) is one major extension of the
CAMx model. PSAT was developed from the related ozone source apportionment method and
provided PM source apportionment for specific geographic regions and source categories (Huang,
Q. et al.,2012). Furthermore, PSAT can be used to analyze the source-acceptor relationship of
$PM_{2.5}$ pollutants, and trace $SO_2$, $SO_4^{2-}$, $NO_3^-$, $NH_4^+$, SOA, Hg, EC, dust particles, and other
primary and secondary particles. As a species tagging method, PSAT tracks the regional source
and industry source of environmental receptor $PM_{2.5}$ and its main chemical components, and then
evaluates the contribution of initial conditions and boundary conditions to PM generation. By
identifying and tracking the transport, diffusion, transformation and decomposion of pollutants
emitted from various sources, PSAT estimates the relative contribution of different emission
sources to the spatial distribution of PM concentrations based on the analysis of mass balance.
PSAT-based source apportionment is conducted using reactive tracers that simulate the nonlinear
transformation between primary PM and secondary PM and are highly efficient and flexible for
source apportionment from the perspective of geographical source regions, emissions source
categories and individual sources (Burr, M. et al.,2011). PSAT effectively avoids the concentration
biases caused by Brute-force based source-closure methods that ignores non-linear chemical
processes and has been widely in previous studies (Xing, J. et al.,2011; Huang, Q. et al.,2012; Wu,
D. et al.,2013; Li, X. et al., 2015; Li, Y. et al., 2015). For this research, we established a regional
transport matrix between pollution sources and environmental receptors. According to the
provincial administrative division, the national grid is divided into 17 divisions, each of which
represents a provincial unit, and all other cells outside the national boundary are classified as Class
I, including the ocean and other areas. According to the scope of the Beijing-Tianjin-Hebei Region
and the "2+26" network, we further divide the study area into 13 sub-divisions, including Beijing,
Tianjin, eight cities in Hebei Province, Henan Province, Shandong Province and Shanxi Province,
for quantifying the influence of local emission and regional transport on the variations in $PM_{2.5}$
concentrations in Beijing during the four pollution episodes.



### 2.4 Model verification

For the mean PM$_{2.5}$ concentrations in Beijing, we compared the observed and model estimated
PM$_{2.5}$ concentrations during the four pollution episodes to verify the accuracy of the WRF-CAMx
model (Fig 2). According to Fig 2, a general agreement was found between the simulated and
observed data with more than 85% of data points falling into the siege area of 1:2 and 2:1 lines.
WRF-CAMx slightly underestimated PM$_{2.5}$ concentrations due to the uncertainty in the emission
inventory, meteorological field simulation errors and insufficient chemical reaction mechanisms.
For the four pollution episodes, the correlation coefficient R, normalized mean bias (NMB),
normalized mean error (NME), mean fractional bias (MFB) and mean fractional error (MFE)
between observed and simulated data ranged from 0.65~0.80, -21%~-10%, 19%~32%, -31%~-8%,
and 12%~34% respectively, indicating a satisfactory simulation output (Boylan et al., 2006).

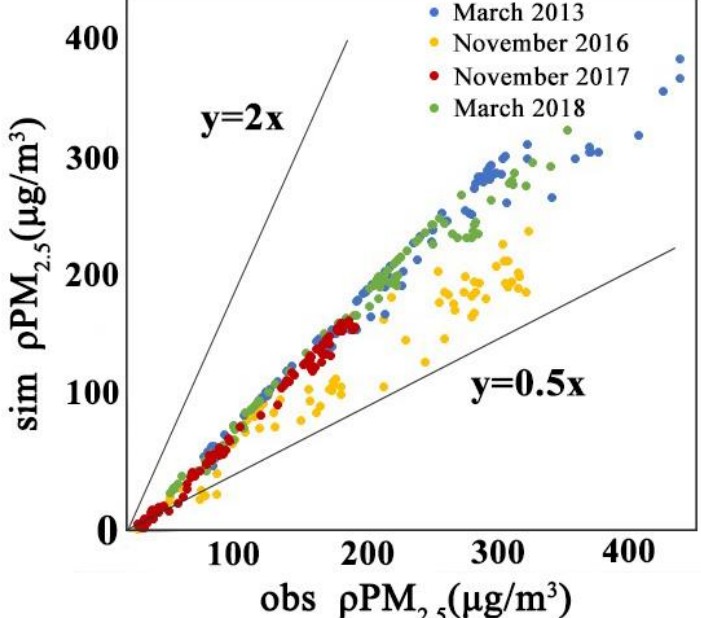

**Fig 2. The comparison of observed and model simulated PM$_{2.5}$ concentrations in Beijing during**
**four pollution episodes.**

### 3 Results

### 3.1 Temporal variations in PM$_{2.5}$ concentrations during the four pollution episodes

Chen et al. (2017, 2018) suggested that wind speed and relative humidity were the major
meteorological factors that influenced wintertime PM$_{2.5}$ concentrations in Beijing. Based on the





ground observation data, we found that the two meteorological factors during the pollution
episode in November 2016 was fairly similar to that during the orange alert period in November
2017, while the meteorological condition during the pollution episode in March 2013 was fairly
similar to that during the orange alert period in March 2018 (as shown in Table 2). According to
Table 2, all the four pollution episodes experienced a high-humidity and weak-wind condition.
Specifically, the fairly high relative humidity for the "2+26" orange alert period in November
2017 and the fairly low wind speed for the "2+26" orange alert period in March 2018 led to
extremely unfavorable conditions for the dispersion of airborne pollutants.
**Table 2 Major meteorological conditions during the four pollution episodes.**

| Pollution Episodes | Mean Relative Humidity(%) | Mean Wind Speed(m/s) |
|---|---|---|
| March, 2013 (No emission-reduction) | 56.25 | 2.32 |
| March, 2018   ("2+26" strategy) | 57.25 | 1.72 |
| November, 2016 (Local emission-reduction) | 58.25 | 2.37 |
| November, 2017 ("2+26" strategy) | 72.25 | 2.09 |

When the meteorological influences on the variations of PM$_{2.5}$ concentrations were limited, a
comparison between the PM$_{2.5}$ concentrations during these two orange alert periods and that
during the two corresponding pollution episodes provides useful reference for evaluating the
effects of "2+26" strategy on PM$_{2.5}$ reductions during the pollution episodes, which are usually
observed under a stagnant atmospheric condition, with high relative humidity and low wind speed.
The temporal variations in PM$_{2.5}$ concentrations during the two "2+26" orange alerts and the two
corresponding pollution episodes are shown in Table 3 and Fig 3.
**Table 3 Characteristics of PM$_{2.5}$ concentrations during four pollution episodes**

| Pollution episode | Mean (µg/m³) | Peak (µg/m³) | Duration of PM$_{2.5}$>100 µg/m³ (h) | Duration of PM$_{2.5}$>150 µg/m³ (h) | Duration of PM$_{2.5}$>200 µg/m³ (h) | Period with PM$_{2.5}$>300 µg/m³ (h) |
|---|---|---|---|---|---|---|
| March, 2013 (No emission-reduction) | 208.49 | 426.12 | 10 | 24 | 37 | 12 |
| March, 2018 ("2+26" strategy) | 174.24 | 325.91 | 6 | 10 | 44 | 5 |
| November, 2016 (Local emission-reduction) | 138.05 | 310.30 | 14 | 9 | 26 | 5 |
| November, 2017 ("2+26" strategy) | 92.91 | 176.20 | 24 | 24 | 0 | 0 |






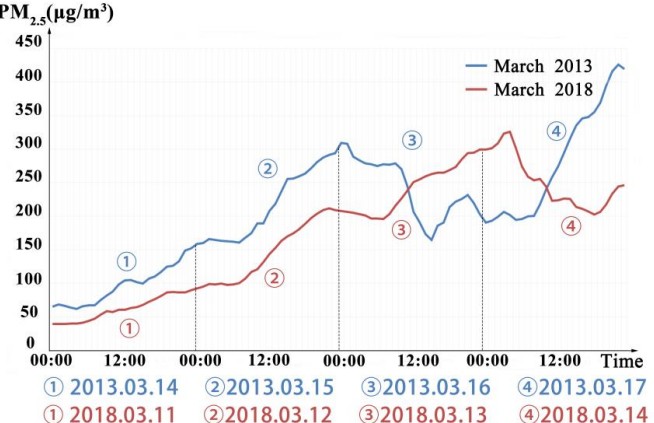

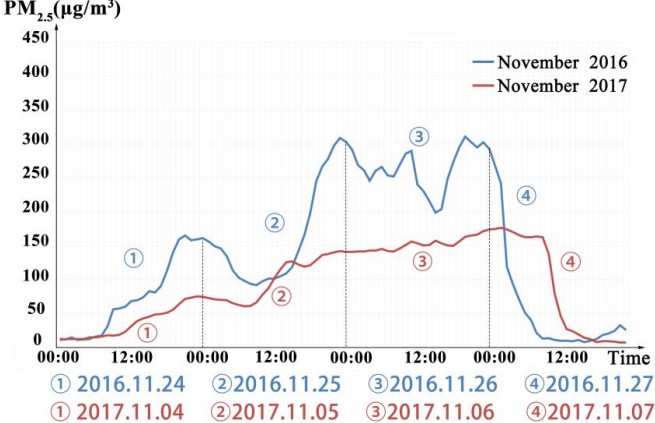

**Fig 3. Variations of PM$_{2.5}$ concentrations during four pollution episodes with different emission-reduction measures in Beijing**

As shown in Table 3 and Figure 3, both the peak and average PM$_{2.5}$ concentrations during the two orange alert periods were remarkably lower than those during the two corresponding pollution episodes with similar initial PM$_{2.5}$ levels and meteorological conditions. For the pollution episode in March 2013 and March 2018, PM$_{2.5}$ concentrations were both around 50μg/ m$^3$ at the beginning of both periods. Similarly, for the pollution episode in November 2016 and November 2017, the initial PM$_{2.5}$ concentrations were both around 15 μg/m$^3$ at the beginning of both periods. Following the similar initial PM$_{2.5}$ concentrations, it is noted that PM$_{2.5}$ concentrations increased at a much lower rate and further led to a lower peak and average PM$_{2.5}$ concentrations during the two orange alert periods.

According to Table 3, the mean and peak PM$_{2.5}$ concentrations during the "2+26" orange alert period in March, 2018 was 16.43% and 23.52% lower than those during the pollution episode in March 2018 respectively. Meanwhile, the duration with extremely high PM$_{2.5}$ concentrations was notably shorter during the orange alert period. The "2+26" strategy implemented during the





orange alert period in November 2017 led to even better effects on PM$_{2.5}$ reductions. The mean
and peak PM$_{2.5}$ concentrations during this period was 32.70% and 43.22% lower than those during
the pollution episode in November 2016 respectively. More importantly, during the entire orange
alert period, PM$_{2.5}$ concentrations were constantly lower than 200 µg/m$^3$, indicating a highly
efficient control of high PM$_{2.5}$ concentrations.
**3.2 PM$_{2.5}$ component analysis during four pollution episodes**
The temporal variations of different PM$_{2.5}$ components during the four pollution episodes are
shown in Fig 4. As the figure indicates, the components of PM$_{2.5}$ in Beijing during the four
pollution episodes had notable variations.





a. The pollution episode in March, 2013 (No emission-reduction)

b. The pollution episode in March, 2018 ("2+26" orange alert period)

c. The pollution episode in November, 2016 (Local emission-reduction)

d. The pollution episode in November, 2017 "2+26" orange alert period)

**Fig 4. The variation of PM$_{2.5}$ components in Beijing during four pollution episodes**

The blank area for the pollution episode in March, 2013 resulted from missing data caused by equipment error





For the four pollution episodes with different emission-reduction measures, the main components
for $PM_{2.5}$ were all $SO_4^{2-}$, $NO_3^-$, and $NH_4^+$. However, some major differences existed. With no or
only local emission-reduction measures implemented, the dominant $PM_{2.5}$ components was $SO_4^{2-}$
for the pollution episode in March 2013 and November 2016. During two "2+26" orange alert
periods, $NO_3^-$ became the dominant $PM_{2.5}$ components. Except for the pollution episode in March
2018, the proportion of another major ion $NH_4^+$ was generally consistent during the four pollution
episodes. The mean mass concentrations and proportions of $SO_4^{2-}$, $NO_3^-$, and $NH_4^+$ during the four
pollution episodes are shown in Table 4.
**Table 4. The mean mass concentration and percent of three major $PM_{2.5}$ components during**
**four pollution episodes**

| Pollution Episode | $SO_4^{2-}$ | | $NO_3^-$ | | $NH_4^+$ | |
|---|---|---|---|---|---|---|
| | Concentration ($\mu g/m^3$) | Percent | Concentration ($\mu g/m^3$) | Percent | Concentration ($\mu g/m^3$) | Percentage |
| **March, 2013** (No emission-reduction) | 45.11 | 39.92% | 34.63 | 30.65% | 27.14 | 24.02% |
| **March, 2018** ("2+26" strategy) | 20.24 | 14.66% | 50.17 | 36.34% | 14.00 | 10.14% |
| **November, 2016** (Local emission-reduction) | 14.96 | 30.10% | 16.19 | 32.57% | 11.66 | 23.45% |
| **November,2017** ("2+26" strategy) | 7.37 | 13.03% | 33.42 | 59.06% | 12.33 | 21.80% |

Through comparison, we found a dramatic decrease of $SO_4^{2-}$ and a notable increase of $NO_3^-$
during two orange alert periods. The main source for $SO_4^{2-}$ is the combustion of fossil fuels
(Shimano, S. et al.,2006; Kuenen, J. et al.,2013), especially the intensive burning of sulfur coals
for wintertime central-heating, manufacturing and household use. The main source for $NO_3^-$ is
vehicle exhaust (Rodríguez, S. et al.,2004; Watson, J. G. et al.,2007; Zeng, F. et al.,2010). $NH_4^+$ is
the secondary pollutant of urban $NH_3$, the main source of which is the decomposition of organic
elements (Frank, D. S. et al.,1980; Watson, J. G. et al.,2007) and the combustion of fossil fuels
(Frank, D. S. et al.,1980; Watson, J. G. et al.,2007; Pan et al., 2016). Through a novel approach,
Pan et al (2016) quantified that more than 90% $NH_3$ in the Beijing-Tianjin-Hebei Region during
heavy pollution episodes resulted from the combustion of fossil fuels. The large variations of
$PM_{2.5}$ components during these episodes was mainly attributed to long-term environmental





policies and contingent emission-reduction measures. A large number of small polluting factories
in Beijing and its surrounding areas have been shut down, and the use of household coal,
especially coarse coal that produces large amounts of sulfate-related pollutants, has been restricted
significantly. In addition to long-term environmental protection policies, contingent
emission-reduction measures, including the temporal shut-down of many factories that consumes
a large amount of coal, were implemented during air pollution alert periods. Furthermore, the
recently launched "2+26" plan requires that areas surrounding Beijing, including many cities in
Hebei Province (e.g., Tangshan) well-known for their coal-based iron industries, should take
simultaneous emission-reduction actions during regional pollution episodes. These long-term and
contingent strategies led to a notable decrease of $SO_4^{2-}$ through local emission-reduction measures
and a further decrease of $SO_4^{2-}$ through "2+26" regional emission-reduction measures. Meanwhile,
the much lower proportion of $NH_4^+$ in March 2018 also indicated a sudden reduction in coal fuel
usage in Beijing and its neighboring areas in 2018, the major driver for which will be explained in
the following section. Conversely, during the four pollution episodes, no strict regulation was
placed on the control of vehicle exhaust. Hence, the notable decrease of $SO_4^{2-}$ and generally
constant mass concentration of $NO_3^-$ led to a rapidly rising proportion of $NO_3^-$ among the $PM_{2.5}$
components during the two orange alerts.

### 3.3 Source apportionment during the four pollution episodes

Based on $PM_{2.5}$ component analysis and PSAT-based source apportionment, we further quantified
the relative contributions of different sources to $PM_{2.5}$ concentrations in Beijing during the four
pollution episodes (Fig 5). A major difference between the pollution episode in March 2013 and in
March 2018 was the dramatic decrease in the relative contribution of coal combustion from 40%
to 11%. Meanwhile, the relative contribution of vehicle exhaust increased significantly from 19%
to 54%, indicating that vehicle exhaust became the dominant source for the pollution episode in
March 2018 with the "2+26" regional emission-reduction measures. On the other hand, the
difference in the relative contributions of different sources between the two pollution episodes in
November 2016 (with local emission-reduction measures) and November 2017 (with "2+26"
regional emission-reduction measures) were much smaller. The major differences lied in the
notable increase in the relative contribution of vehicle exhaust from 29% to 40% and the decrease
in the relative contribution of coal combustion from 34% to 28%.
As described above, the continuous decrease in the relative contribution of coal combustion from
the pollution episodes in 2013 to the episode in 2018 resulted from the combination of long-term





and contingent local and regional emission-reduction measures. Note that despite a similar "2+26"
strategy implemented, the relative contribution of coal combustion during the orange alert period
in November 2017 was much higher than that in March 2018. A major reason for this dramatic
change in a short period was the implementation of a large-scale environmental project. Before
November 2017, the starting point of central heating in Beijing, a regional project called "Coal to
Gas" had finished replacing coal-based central heating systems by gas-based systems for 1.9
million households in the Beijing-Tianjin-Hebei Region, leading to a 2 million-ton decrease in
coal consumption in the region. As a result, the relative contribution of coal combustion, which
was the dominant emission source for $PM_{2.5}$ in Beijing during the central-heating season from
November to March, decreased to a fairly low level during the orange alert period in March 2018.

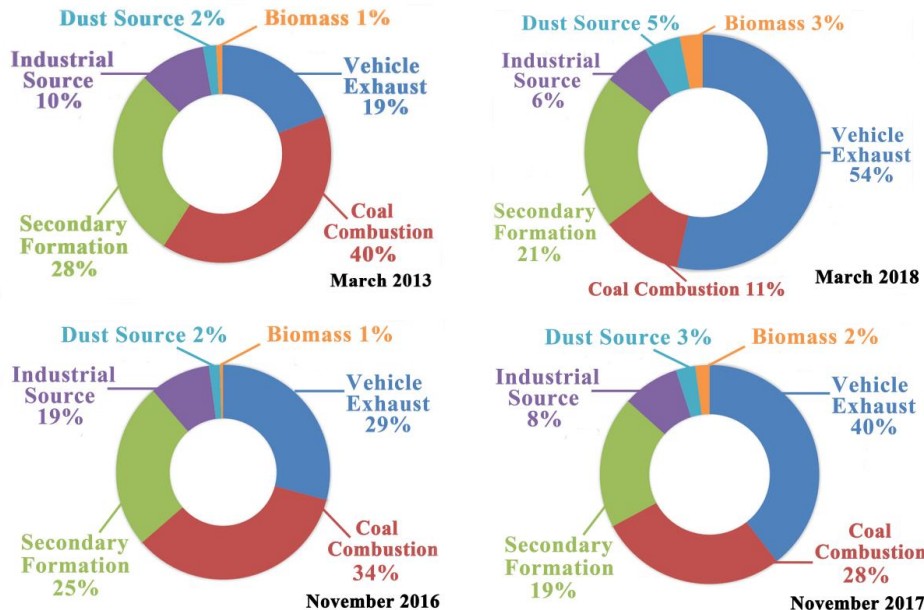

**Fig 5. The relative contribution of different sources to $PM_{2.5}$ concentrations in Beijing**
**during four pollution episodes**
**3.4 The relative contribution of local emission and regional transport to $PM_{2.5}$**
**concentrations in Beijing during the four pollution episodes**
Through the simulation of the WRF-CAMx model based on local and regional emission
inventories, we quantified the relative contributions of local emission and regional transport of
airborne pollutants to the variations in $PM_{2.5}$ concentrations in Beijing during four pollution
episodes (Fig 6). According to Fig 6, the relative contributions of local emission and regional



transport to PM$_{2.5}$ concentrations in Beijing varied notably. For the pollution episode in March
2013 with no emission-reduction measures, the relative contribution of local emissions was
69.27%, much lower than the 88.35% for the "2+26" orange alert period in March 2018. On the
other hand, for the pollution episode in November 2016 with local emission-reduction measures,
the relative contribution of local emissions was 76.83%, much higher than the 49.46% for the
"2+26" orange alert period in November 2017. Meanwhile, the relative contribution to PM$_{2.5}$
concentrations in Beijing from specific areas also differed significantly during these pollution
episodes. We found that different emission-reduction strategies did not lead to a clear pattern for
the relative contributions of local emission and regional transport. One major reason for this is that
the regional transport of airborne pollutants from neighboring areas to Beijing is influenced
significantly by meteorological conditions, the intensity of regional emission sources and the
regional distribution of PM$_{2.5}$ concentrations, which demonstrated remarkable seasonal variations
and synoptic-scale uncertainties. From this perspective, we attempted to explain the underlying
drivers for the variations in local and regional contributions to PM$_{2.5}$ concentrations during the
four pollution episodes.

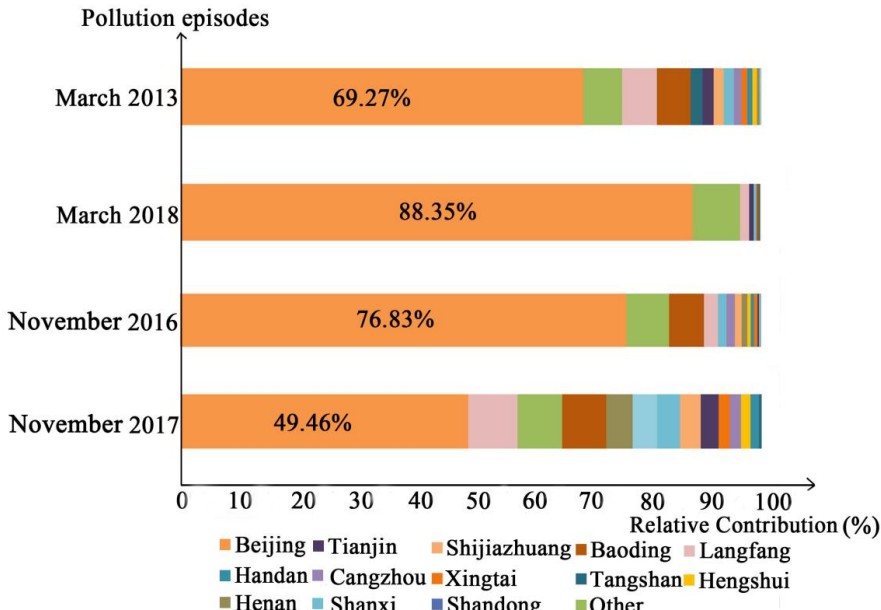


**Fig 6. The relative contributions of local emission and regional transport to PM$_{2.5}$**
**concentrations in Beijing during the four pollution episodes**
For the pollution episode in March 2013, without long-term and contingent emission-reduction
measures, the large amount of combusted coal fuels in the neighboring areas of Beijing led to a



relatively large regional contribution. For the pollution episode in March 2018, with the
implementation of the large-scale "Coal to Gas" project and "2+26" strategy, the rapidly reduced
coal consumption in cities surrounding Beijing and the limited restriction on the emission of local
vehicles led to a fairly high local contribution. For the pollution episode in November 2016, an
inversed temperature layer was observed with high relative humidity, which was a favorable
environment for the production of secondary $PM_{2.5}$ and a relatively large local contribution.
Despite the implementation of the "2+26" strategy, the abnormally high regional contribution for
the pollution episode in November 2017 could be attributed to the prevailing southerly winds that
brought in a large amount of air from neighboring cities (e.g., Shijiazhuang). Therefore, although
this pollution episode occurred in winter, it had more similarities to a summertime pollution
episode, which was characterized by prevailing southerly winds, thoroughly mixed pollutants
within the Beijing-Tianjin-Hebei Region, and notable regional transport.

## 4 Discussion

Through the comparison of the components of $PM_{2.5}$ during the pollution episode in 2013 and
those in 2016, 2017 and 2018, we found that the proportion of sulfate ions decreased significantly
while nitrite ions became the dominant component of $PM_{2.5}$ during the pollution episodes. This
result is consistent with findings from some recent studies (Fromme H et al., 2008; Tan J et al.,
2016; Shang X et al., 2018). The dramatic decrease in the proportions of $SO_4^{2-}$ and $NH_4^+$ ions in
$PM_{2.5}$ during the pollution episode in March 2018 suggested a satisfactory effect of long-term
pollution control measures, especially the "Coal to Gas" project, and contingent "2+26" regional
emission-reduction measures on the reduction of coal combustion in the Beijing-Tianjin-Hebei
Region. As revealed by previous studies (Zhang R et al., 2013; Liu Y et al., 2018; Shang X et al.,
2018) and the source apportionment from this research, the use of coal fuels has been the
dominant source for the formation and mass concentration of $PM_{2.5}$ in Beijing since 2013.
However, the remarkable decrease in coal combustion since the winter of 2017 has greatly
reduced the contribution of coal combustion to local $PM_{2.5}$ concentrations, which directly
improved the wintertime air quality and led to the cleanest winter in Beijing since 2013. The mean
wintertime (the winter for Beijing here refers to the central-heating season from November 15th to
March 15th) $PM_{2.5}$ concentration in Beijing for 2013, 2014, 2015, 2016 and 2017 was 88.19,84.41,
89.39, 92.39 and 47.31 μg/ m³ respectively.
The implementation of the "2+26" strategy led to different effects on $PM_{2.5}$ reductions during



specific pollution episodes. In addition to different emission-reduction strategies, the improvement
of air quality in Beijing is controlled by a diversity of factors. First, meteorological conditions
exert a strong influence on the accumulation and dispersion of local airborne pollutants in Beijing
and the long-distance transport of airborne pollutants from neighboring areas. Second, the
distribution of $PM_{2.5}$ concentrations in the "2+26" region determines whether the air brought into
Beijing from neighboring areas increases or decreases $PM_{2.5}$ concentrations there. Third, the $PM_{2.5}$
level during pollution episodes influences the relative contributions of local and regional
contributions. The mean $PM_{2.5}$ concentrations during the "2+26" orange alert period in March
2018 was 174.24 $\mu g/m^3$. High-concentration $PM_{2.5}$ during pollution episodes led to a stagnant
condition with high humidity and low wind speed (Chen et al. 2017, 2018), which was an
unfavorable condition for the regional transport of airborne pollutants. Therefore, the relative
contribution of local emission to this extremely high $PM_{2.5}$ concentrations was 88.35% while the
relative contribution of regional transport was 11.65%. In this case, although unified
emission-reduction measures were implemented in its neighboring areas, the significantly
restricted regional transport did not fully project the effects of the "2+26" strategy to the local
$PM_{2.5}$ concentrations in Beijing. Conversely, the mean $PM_{2.5}$ concentrations during the "2+26"
orange alert period in November 2017 was 92.91 $\mu g/m^3$, which was not high enough to
significantly prevent the regional transport of airborne pollutants. Therefore, the "2+26" strategy
led to a simultaneous reduction in $PM_{2.5}$ concentrations in this region and a large amount of clean
air from its neighboring cities that significantly diluted the local $PM_{2.5}$ in Beijing. Consequently,
the relative contribution of regional transport was larger than 50% and thus the "2+26" strategy
achieved a much better effect on $PM_{2.5}$ reductions than that in March 2018.
Another dominant factor that influences the effects of the "2+26" strategy is the level of air
pollution alert and its corresponding emission-reduction measures. With the launch of orange air
pollution alerts, a series of restrictions are placed on the temporary shut down of polluting
factories and the emission of fossil fuels can be reduced significantly. However, during orange
alert periods, only the use of a small proportion of vehicles that cannot meet Environmental
Standards Level I and II are forbidden whilst no additional regulation is implemented on the use
of more than 5 million private cars in Beijing. As a result, the relative contribution of vehicle
exhaust increased rapidly during two of the "2+26" orange alert periods. Especially for the orange
alert period in March 2018, vehicle exhaust contributed to more than 50% of the high $PM_{2.5}$
concentrations that were higher than 174.24 $\mu g/m^3$. With dramatically reduced use of coal fuels in



the Beijing-Tianjin-Hebei Region due to the recent completion of the "Coal to Gas" project, the
control of vehicle exhaust is increasingly crucial for managing $PM_{2.5}$ concentrations during
pollution episodes. In this light, red air pollution alerts, which have stricter regulations on the use
of vehicles, should be employed with the "2+26" regional emission-reduction strategy during
heavy pollution episodes. For instance, during the heavy pollution episode in March 2018, if a red
alert instead of the orange alert was issued, the implementation of odd–even license plate policy
would instantly cut the daily use of private cars in Beijing by fifty percent and significantly reduce
the contribution of vehicle exhaust to $PM_{2.5}$ concentrations. Given the growing contribution of
vehicle exhaust to $PM_{2.5}$ pollutions in Beijing, in addition to the contingent regulations during
pollution episodes, long-term policies, including the improvement of the public transit system, the
enhancement of petrol quality and promotion of electric cars, should be properly implemented for
further reducing vehicle-exhaust induced $PM_{2.5}$ pollutions.
Although the regional transport network for air pollution in Beijing has been identified, this
research suggested that only those cities adjacent to Beijing, such as Baoding, Shijiazhuang and
Lang fang, made a relatively large contribution to the $PM_{2.5}$ concentrations in Beijing whilst the
relative contributions of some other areas within the "2+26" framework were very limited.
Considering the substantial social and economic loss induced by the implementation of air
pollution alerts, city-specific, rather than region-wide unified emission-reduction strategies,
should be conducted for promoting air quality in Beijing during pollution episodes. Tight
measures can be implemented in cities that make large contributions while lenient measures can
be implemented in cities that make limited contributions to $PM_{2.5}$ concentrations. To this end,
future studies should place more emphasis on quantifying the relative contributions from different
cities to local $PM_{2.5}$ concentrations in Beijing and setting city-specific emission-reduction
measures for each city within the "2+26" region.

## 5 Conclusions

We compared the variations in $PM_{2.5}$ concentrations in Beijing during four recent pollution
episodes with different emission-reduction strategies. Based on this comparison, we found that the
"2+26" regional emission-reduction strategy implemented in March 2018 led to a mean $PM_{2.5}$
concentrations of only 16.43% lower than that during the pollution episode in March 2013, when
no emission-reduction measure was in place. On the other hand, the same "2+26" strategy
implemented in November 2017 led to a mean $PM_{2.5}$ concentrations of 32.70% lower than that



during the pollution episode in November 2016 with local emission-reduction measures. The
result suggested that the effects of the "2+26" regional emission-reduction measures on PM$_{2.5}$
reductions were influenced by meteorological conditions, regional distribution of PM$_{2.5}$
concentrations and local PM$_{2.5}$ level, and could differ significantly during specific pollution
episodes. Based on our PM$_{2.5}$ component analysis, we found that the proportion of sulfate ions
decreased significantly and nitrate ions were the dominant PM$_{2.5}$ components during the two
"2+26" orange alert periods. The source apportionment revealed that the relative contribution of
coal combustion to PM$_{2.5}$ concentrations during the pollution period in March 2013, November
2016, November 2017 and March 2018 was 40%, 34%, 28% and 11% respectively, indicating that
the recent completion of the large-scale "Coal to Gas" project and contingent "2+26" regional
emission-reduction measures led to a dramatic decrease in coal combustion in the
Beijing-Tianjin-Hebei Region. Meanwhile, with no specific regulation on the use of private cars,
the relative contribution of vehicle exhaust during the "2+26" orange alert periods in November
2017 and March 2018, was 40% and 54% respectively. The relative contribution of local
emissions to PM$_{2.5}$ concentrations in Beijing varied significantly and ranged from 49.46% to
89.35% during the four pollution episodes. With gradually reduced coal consumption in the
Beijing-Tianjin-Hebei region, this research suggested that the "2+26" regional emission-reduction
strategy should be implemented with red air pollution alerts to intendedly reduce the dominant
contribution of vehicle exhausts to PM$_{2.5}$ concentrations. Meanwhile, emission-reduction policies
should be designed and implemented accordingly for different cities within the "2+26" regional
framework. The methodology and findings from this research provided useful reference for
comprehensively understanding the effects of the "2+26" strategy, and for better implementation
of future long-term and contingent emission-reduction measures during heavy pollution episodes.

## Acknowledgement

Sincere gratitude goes to Tsinghua University, which produced the Multi-resolution Emission
Inventory for China (http://meicmodel.org/) and Research center for air quality simulation and
forecast, Chinese Research Academy of Environmental Sciences (http://106.38.83.6/), which
supported the model simulation in this research. This research is supported by State Key
Laboratory of Earth Surface Processes and Resource Ecology (2017-KF-22), National Natural
Science Foundation of China (Grant Nos. 41601447), the National Key Research and
Development Program of China (NO.2016YFA0600104), and Beijing Training Support Project
for excellent scholars (2015000020124G059).



**Author contribution**


Chen, Z., Xu, B and Yang, L. designed this research. Chen, Z wrote this manuscript. Chen, D.,
Zhuang, Y, Cheng, N, Gao, B. Li, R. and Zhao, B conducted data analysis. Chen, D and Zhuang, Y.
produced the figures. Kwan, M., Yang, L. and Chen, B helped revise this manuscript.



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
