# Peer review of "Evaluating the "2+26" Regional Strategy for Air Quality"

_Atmospheric Chemistry and Physics, 2018_

## Referee Comment (RC1) · Anonymous Referee #1 · 21 Dec 2018

General comments: This manuscript attempts to evaluated the effects of "2+26" regional integrative strategy on air quality improvement in China, by comparing of variation of PM2.5 concentrations and source contribution in Beijing during four pollution episodes. The study uses airborne pollutants modeled by the WRF-CAMx model and the observed PM2.5 to understand how "2+26" strategy can affect the PM2.5 reduc-

tion during the pollution episodes. The topic is interested ones because the impact of emission-reduction on air quality has been still unclarified, and the results was very helpful to control the air pollution in cities. However, the authors have not made best use of the model results. The model can provide information on meteorological condition, emission and regional distribution of pollutants. These information should be supplemented and analyzed to know the influence of meteorological condition on the implementation of "2+26" strategy. There needs to be significant improvement in results interpretation and more analysis needs to be done to know the control factors of variation of PM2.5 and SNA. I would thus recommend a major revision to improve this manuscript.

Specific comments: Q1. In Discussion Session, the authors stated the influence of meteorological factors on PM2.5 concentration and explained the different effects of "2+26" strategy on PM2.5 reduction for different pollution episodes. Using WRF-CAMx model, author can obtain the detailed meteorological field. However, only average RH and wind speed for four pollution episodes was shown in Table 2. Then how authors determine the airborne pollutants in Beijing was transported from neighboring areas during pollution episodes? There is also no meteorological data under unpolluted weathers (AQI < 50), how can we know the meteorological difference between pollution and clean days? I can't follow "a high-humidity condition" without comparison with a background value. In Line 438-444, the author pointed that "although unified emission-reduction measures were implemented in its neighboring areas, the significantly restricted regional transport did not fully project the effect of the "2+26" strategy to the local PM2.5 concentrations in Beijing", so to what extent can meteorological condition affect the implementation of "2+26" strategy? And under what circumstances the meteorological condition will have important effect on implementation? Q2. There is no detailed information of meteorological parameters and concentration of SO2, NOx, and NH3 during four pollution episodes. However, the SNA formation is related with the precursor. I think the related information should be supplemented and analyzed before comparison of PM2.5 reduction. Q3. In Introduction Section, the authors explained the

specific emission-reduction measures in detail. However, there is no emission data for airborne pollutants, such as SO2, NOx, NH3, dust, etc., from 2013 to 2018. Whether the emission of all these precursor gas was really reduced greatly by implementing "2+26" strategy? I suggested that the authors analyze the emission variation in detail to evaluate the "2+26" strategy. Q4. Section 2.2.2. Supplement the sampling duration, sample numbers and the membrane to collected PM2.5. Was the sampling duration 15min for ions and 1 hour for OC/EC? Q5. Revise the Section 2.3.1 to make the description of WRF-CAMx more concise Q6. Section 2.3.1 In this manuscript, the authors simulated several episodes during 2013-2017. During these years, emissions in China changed obviously due to lots of national strategies. Emission inventory is an important factor which would influence model results. So please clarify which years' emission inventories were used in this study? Did you consider "coal to gas" strategy in your emission inventory? Q7. L207-209. The input and output of CMAx is in binary format. However, output from MCIP is in NETCDF format, please clarify how to use NETCDF meteorological data in CMAx? Q8. Section 2.4 The authors explained that meteorological parameters contributed to the underprediction of simulated PM2.5, could you give out some information about the model performance of meteorological parameters such as T, RH, WS, WD? Q9. The author found that composition of PM2.5 changed obviously due to the national strategies, therefore it is important to show the model performance of inorganic components in PM2.5 such as SO42-, NO3-, NH4+ but not only show the result of PM2.5. If the model performance is satisfied, further analysis of PSAT would be reasonable, otherwise, results of PSAT would not be convincing. Q10. Supplement the criteria or error index that can verify the satisfactory simulation for PSAT. Q11. In Section 3.2, only the variation of ions in PM2.5 was discussed. Organic compounds are one of major components of PM2.5. Since the OC/EC has be analyzed, I think the OC variation should be discussed here. Q12. From Fig. 4b, very high concentration of NO2- was observed during the pollution episode in March, 2018. The value is very abnormal, almost two times higher than NO3-. In general, nitrite shows very low concentration in atmospheric aerosols and contributes little to

water soluble inorganic ions. What's the reason for this abnormal value? I think the authors should check the data and discussed the reason. Q13. L320-321. Which data can support the "The main source for NO3- is vehicle exhaust" in Beijing? How did you verified the vehicle exhaust was main sources of NO3- in Being just as the cited reference suggested in other cities? I think the source appointment of NO3- will be helpful to support your suggestion on vehicle exhaust in conclusion. Q14. L320-321. As "The main source for NO3- is vehicle exhaust" and the vehicles that cannot meet the Environmental Levels I and II was forbidden during orange alerts, why the concentration of NO3- was much higher during orange alerts in Mar, 2018 than that in March, 2013 without emission-reduction (Table 4)? Increased NO3- corresponded to deceased concentration of NH4+ during pollution episodes, so what's possible existing form of NO3- in PM2.5? Q15. L364-365. Please clarify what changes have been made to the air pollutants emission after "Coal to Gas". Q16. L378-383. According to Fig.6, the local emission contributed 49.46% - 88.35% to PM2.5 during four pollution episode, indicating the local emission had a great effect on PM2.5 in Beijing. This is contradicting L92-93. And the different emission-reduction strategies did not lead to a clear pattern for the regional transport. So the PM2.5 reduction really was a result of "2+26" strategies or the meteorological condition? Whether the strict regulation on vehicle exhaust will be more effective than that of regional emission control under specific wind direction? The meteorological condition should be analyzed in detail for each pollution episode. Q17. Overall, some of the conclusions on page 20 appear to be speculation with little data or discussion to support it, such as L494-495. Analysis and discussion on regional distribution of PM2.5 needs to be supplemented.

Technical corrections: L139. Supplement the link of website PM25.in. It's difficult to follow. L164. Change "ãĂĄ" to ",". L183. Change "*" to "ïĆť". P12. Fig.4, change "NO2-" to "NO2-"

Please also note the supplement to this comment:
https://www.atmos-chem-phys-discuss.net/acp-2018-1085/acp-2018-1085-RC1-

supplement.pdf

---

## Referee Comment (RC2) · Anonymous Referee #2 · 18 Feb 2019

This manuscript analyzed four pollution episodes and evaluated the effects of "2+26" regional integrative strategy on air quality improvement in Beijing. Observation and model simulation were used to investigate how the emission reduction influence the pollution episodes. In general, this is an interesting topic and helpful for the Chinese policy maker to improve the pollution management. However, the analysis in

this manuscript was not convincing enough, and the model results were not used well. I recommend a major revision before can be published on ACP.

General comments:

1. It is dangerous to evaluate the pollution control strategy by using only four pollution episodes. Too many parameters, especially the meteorological parameters, can influence the pollution level in one case, and would result in large uncertainties in the evaluation. A comparison based a long-period observation is needed. The current comparison between every two episodes, at least, not statistical significant.

2. In Fig. 2, it seemed to me the simulation result was too good. And the model can only underestimate PM2.5, but not overestimate, why? The author need to provide the comparison results of the chemical composition, but not only the mass concentration of PM2.5.

Specific comments:

1. Remove "recent" in the title

2. I would not recommend use 'Orange air pollution alert' in the title.

3. In Fig.3, this kind of direct comparison between two cases at different time did not make much sense.

4. In Fig. 4b, why there were such a high concentration of nitrite and chloride?

5. In Fig. 5, compared to the previous pollution episodes, the contribution of coal combustion in March 2018 episode decreased a lot, but the November 2017 case did not, why?

---

## Author Comment (AC1) · 14 Apr 2019

**To Reviewer 1:**

Thanks so much for your valuable comments, which improved this manuscript a lot. We have fully revised this manuscript according to your general and detailed comments. More description concerning the model setting and  $PM_{2.5}$  component was added to the revised manuscript and some required explanations were included as well. We are more than happy to conduct further revisions if additional requirements are given.

Q1. In Discussion Session, the authors stated the influence of meteorological factors on PM2.5 concentration and explained the different effects of "2+26" strategy on PM2.5 reduction for different pollution episodes. Using WRF-CAMx model, author can obtain the detailed meteorological field. However, only average RH and wind speed for four pollution episodes was shown in Table 2. Then how authors determine the airborne pollutants in Beijing was transported from neighboring areas during pollution episodes? There is also no meteorological data under unpolluted weathers (AQI < 50), how can we know the meteorological difference between pollution and clean days? I can't follow "a high-humidity condition" without comparison with a background value. In Line 438-444, the author pointed that "although unified emission-reduction measures were implemented in its neighboring areas, the significantly restricted regional transport did not fully project the effect of the "2+26" strategy to the local PM2.5 concentrations in Beijing", so to what extent can meteorological condition affect the implementation of "2+26" strategy? And under what circumstances the meteorological condition will have important effect on implementation?

R: Thanks for this question. We are sorry that we did not make this clear in the previous manuscript. In this research, we employed the default meteorological field, including dozes of meteorological factors, for running WRF-CAMx during the four pollution episodes and thus the comprehensive meteorological influences on the transport of airborne pollutants in the "2+26" region has been considered. In table, we simply listed the ground observed RH and wind speed were listed in the table for the following reason. Firstly, the accuracy of all those meteorological factors provided by the WRF-CAMx model were not sure, as we simply got the ground observation meteorological data including several major meteorological factors for comparison. Therefore, we listed the ground-observed "RH" and "wind speed" with high accuracy to demonstrate the major meteorological conditions for two corresponding episodes were similar. Secondly, our previous studies (Chen et al., 2016,2017,

2018) proved that wind speed and relative humidity were the dominant meteorological factors for  $PM_{2.5}$  concentrations in Beijing and exerted a major influence on the concentration and dilution of  $PM_{2.5}$  in Beijing during the period from November to March. Similarly, according to a recent formal governmental report based on a series of studies (https://m.21jingji.com/article/20190311/herald/263828cd8f4cf3986ee1c39378c64881.html?fr om=groupmessage&isappinstalled=0), compared with normal meteorological conditions, high humidity (around 60%) and low wind speed (around 2m) were unfavorable meteorological conditions for the dispersion of  $PM_{2.5}$  and could lead to haze episodes in the Beijing-Tianjin-Hebei region. Therefore, here we presented the ground observed RH and wind speed value, which had a much high accuracy than WRF-CAMx provided meteorological field data, to demonstrate the major meteorological conditions that influence regional transport of airborne pollutants were similar during each pair of corresponding pollution episodes and thus the different trend of  $PM_{2.5}$  concentrations during these four pollution episodes were mainly induced by different emission-reduction measures.

As a species tagging method, PSAT tracks the regional source and industry source of environmental receptor  $PM_{2.5}$  and its main chemical components, and then evaluates the contribution of initial conditions and boundary conditions to PM generation. Therefore, the use of WRF-PSAT model can determine "the airborne pollutants in Beijing was transported from neighboring areas during pollution episodes".

In addition, as per your request, we ran some simulations to compare the model simulated meteorological factors with observed values and the results were presented in the revised manuscript.

- Chen, Z.Y., Xie, X., Cai, J., Chen, D., Gao, B., He, B., Cheng, N., Xu, B. 2018. Understanding meteorological influences on PM2.5 concentrations across China: a temporal and spatial perspective, Atmos. Chem. Phys., 18, 5343-5358
- Chen , Z.Y., Xu, B., Cai , J., Gao, B.B. 2016. Understanding temporal patterns and characteristics of air quality in Beijing: A local and regional perspective. Atmospheric Environment. 127, 303-315.

3. Chen, Z. Y., Cai, J., Gao, B. B., Xu, B., Dai, S., He, B., Xie, X. M. 2017. Detecting the causality influence of individual meteorological factors on local PM2:5 concentrations in the Jing-Jin-Ji region, Scientific Reports, 7, 40735.

Q2. There is no detailed information of meteorological parameters and concentration of SO2, NOx, and NH3 during four pollution episodes. However, the SNA formation is related with the precursor. I think the related information should be supplemented and analyzed before comparison of PM2.5 reduction.

Q3. In Introduction Section, the authors explained the specific emission-reduction measures in detail. However, there is no emission data for airborne pollutants, such as SO2, NOx, NH3, dust, etc., from 2013 to 2018. Whether the emission of all these precursor gas was really reduced greatly by implementing "2+26" strategy? I suggested that the authors analyze the emission variation in detail to evaluate the "2+26" strategy.

R: Thanks so much for pointing this out. This is a very good question. The use of meteorological factors were explained as above. The SNA was related to precursors. The major PM2.5 component was SO2 and NO2, which exerted strong influence on the generation of sulfate ion and nitride ion. For this research, we obtained some PM2.5 component data during the four pollution episodes. Since we did not have the NH3 data, we listed the SO2 concentrations in the revised manuscript. We can see, the mean SO2 concentration for March, 2013 was notably higher than that in March 2018 whilst the mean SO2 concentration for November, 2016 were notably higher than those in November, 2017. This is mainly attributed to the fact that during the two "2+26" periods, a large number of factories in the "2+26" region were temporally shut down and thus the coal-combustion induced SO2 concentrations reduced significantly, compared with the corresponding pollution episodes with no or local emission-reduction measures. However, compared with SO2, since no strict regulation on vehicle uses was implemented during the "2+26" orange alert period, the magnitude of NO2 reduction during the pollution episodes was much smaller. And this is the reason, in the discussion part, we suggested that the "red alerts", which suggested the restriction of half vehicles, should be implemented during heavy pollution episodes.

| Pollution Episodes                               | Mean SO 1 ( µg/m 3 ) | Mean NO 2 ( µg/m 3 ) |
|--------------------------------------------------|--------------------------------------------|--------------------------------------------|
| 2013.03.14-2013.03.17 (No emission-reduction)    | 65.25                                      | 98.25                                      |
| 2018.03.11-2018.03.14 ("2+26" strategy)          | 14.25                                      | 76                                         |
| 2016.11.24-2016.11.27 (Local emission-reduction) | 17.75                                      | 82.25                                      |
| 2017.11.04-2017.11.07 ("2+26" strategy)          | 4.58                                       | 60.25                                      |

Q4. Section 2.2.2. Supplement the sampling duration, sample numbers and the membrane to collected PM2.5. Was the sampling duration 15min for ions and 1 hour for OC/EC?

R: Yes, We collected the data at the Dongsi station and the sampling duration for ions was 15 min and OC/EC was 1 hour using automatic URG-9000B Ambient Ion monitor (Thermo Fisher Scientific), which includes two Dionex ICS-90 ion chromatography systems (DIONEX, US). The membrane used for this automatic ambient ion monitor was denuder, which realized the separation of particulate matters and gas by absorbing gas using liquid.

Q5. Revise the Section 2.3.1 to make the description of WRF-CAMx more concise

R: Thanks for this comment. We have shortened the part accordingly in the revised manuscript.

Q6. Section 2.3.1 In this manuscript, the authors simulated several episodes during 2013-2017. During these years, emissions in China changed obviously due to lots of national strategies. Emission inventory is an important factor which would influence model results. So please clarify which years' emission inventories were used in this study? Did you consider "coal to gas" strategy in your emission inventory?

R: The emission inventory was updated every year and we employed corresponding emission inventory for each pollution episode. For the pollution episode in March, 2018, since the emission inventory in 2017 has yet been updated, we still employed an updated 2017 emission inventories. However, the complete inventory included one category of residential emissions. For simulating this episode with the general completion of "coal to gas" project, we reduced the coal-combustion induced emissions (mainly SO2) and increased the gas induced emissions (mainly NO2) according to some general proportions given by official documents. Q7. L207-209. The input and output of CMAx is in binary format. However, output from MCIP is in NETCDF format, please clarify how to use NETCDF meteorological data in CMAx?

R: This is a good question. We employed the camx2WRF module to transfer NETCDF data from WRF to readable data for CAMx. This detail has been added to the revised manuscript.

Q8. Section 2.4 The authors explained that meteorological parameters contributed to the under prediction of simulated PM2.5, could you give out some information about the model performance of meteorological parameters such as T, RH, WS, WD?

R: Thanks so much for this comment. In the revised manuscript, we have added a comprehensive simulation of major meteorological parameters, temperature, relative humidity and wind speed, as well as the simulation of  $PM_{2.5}$  component.

Q9. The author found that composition of PM2.5 changed obviously due to the national strategies, therefore it is important to show the model performance of inorganic components in PM2.5 such as SO42-, NO3-, NH4+ but not only show the result of PM2.5. If the model performance is satisfied, further analysis of PSAT would be reasonable, otherwise, results of PSAT would not be convincing.

R: Thanks so much for this comment. Follow your suggestions, we presented the simulation results of major  $PM_{2.5}$  component  $SO4^{2-}$ ,  $NO^{3-}$ ,  $NH^{4+}$  in the revised manuscript to demonstrate the model performance.

Q10. Supplement the criteria or error index that can verify the satisfactory simulation for PSAT.

**R** : This is a very good point. However, since there is no reference data for the relative contribution of different sources to  $PM_{2.5}$  concentrations, it is highly difficult, if not possible, to verify the accuracy for PSAT or other source-apportionment models. PSAT model was a fixed model, and has been widely in a diversity of studies (Yarwood et al., 2007; Baker and Foley, 2011., Li et al., 2015; Ju H et al., 2018; Zhang et al., 2018;). Most studies directly employed the default setting of PSAT and the simulation results of PSAT were widely accepted as reasonable simulation of source apportionment.

Ju, H., Bae, C., Kim, B. U., Kim, H. C., Yoo, C., & Kim, S. (2018). PM2. 5 Source

Apportionment Analysis to Investigate Contributions of the Major Source Areas in the Southeastern Region of South Korea. JOURNAL OF KOREAN SOCIETY FOR ATMOSPHERIC ENVIRONMENT, 34(4), 517-533.

Zhang, Y., Li, X., Nie, T., Qi, J., Chen, J., & Wu, Q. (2018). Source apportionment of PM2. 5 pollution in the central six districts of Beijing, China. Journal of Cleaner Production, 174, 661-669.

Baker, K. R., & Foley, K. M. (2011). A nonlinear regression model estimating single source concentrations of primary and secondarily formed PM2. 5. Atmospheric Environment, 45(22), 3758-3767.

Yarwood, G., Morris, R. E., & Wilson, G. M. (2007). Particulate matter source apportionment technology (PSAT) in the CAMx photochemical grid model. In Air Pollution Modeling and Its Application XVII (pp. 478-492). Springer, Boston, MA.

Li, X., Zhang, Q., Zhang, Y., Zheng, B., Wang, K., Chen, Y., ... & He, K. (2015). Source contributions of urban PM2. 5 in the Beijing–Tianjin–Hebei region: Changes between 2006 and 2013 and relative impacts of emissions and meteorology. Atmospheric Environment, 123, 229-239.)

Q11. In Section 3.2, only the variation of ions in PM2.5 was discussed. Organic compounds are one of major components of PM2.5. Since the OC/EC has been analyzed, I think the OC variation should be discussed here.

R: Thanks so much for this good comment. According to your comment, we added the  $PM_{2.5}$  component OC and EC, which were collected during these pollution episodes (except for the pollution episode in 2013, when OC and EC data were not collected then), to Fig 4 in the revised manuscript.

Q12. From Fig. 4b, very high concentration of NO2- was observed during the pollution episode in March,2018. The value is very abnormal, almost two times higher than NO3-. In general, nitrite shows very low concentration in atmospheric aerosols and contributes little to water soluble inorganic ions. What's the reason for this abnormal value? I think the authors should check the data and discussed the reason.

R: R: Thanks for pointing this out. Due to data recording errors, the  $NO_3^-$  in the previous manuscript was wrongly used as the nitrite. We corrected this and added additional OC and EC to the revised manuscript. The updated figure was listed as follows. Thanks again for

**pointing this out and we are very sorry for this confusion.**

Q13. L320-321. Which data can support the "The main source for NO3- is vehicle exhaust" in Beijing? How did you verified the vehicle exhaust was main sources of NO3- in Being just as the cited reference suggested in other cities? I think the source appointment of NO3- will be helpful to support your suggestion on vehicle exhaust in conclusion.

R : In addition to the PNAS paper suggested that the main source for  $NO_3^-$  is vehicle exhaust, we cited the official report on the source apportionment of  $PM_{2.5}$  in Beijing (http://www.gov.cn/xinwen/2014-10/31/content\_2773436.htm), which stated that the main source for  $NO_3^-$  was vehicle exhaust. In addition to the source apportionment of  $PM_{2.5}$  in other areas, Han et al., (2007) conducted field survey and also suggested that the main source of  $NO_3^-$  was vehicle exhaust.

Han, L., Zhuang, G., Cheng, S., Wang, Y., & Li, J. . (2007). Characteristics of re-suspended road dust and its impact on the atmospheric environment in beijing. Atmospheric Environment, 41(35), 7485-7499.

On the other side, current source apportionment methods mainly concerned the contribution of different precursors or sources to general PM2.5 concentrations, whilst the capability for source apportionment of individual ions was limited. (Zhang, R., Jing, J., Tao,

J., Hsu, S. C., Wang, G., Cao, J., ... & Shen, Z. 2013. Chemical characterization and source apportionment of PM 2.5 in Beijing: seasonal perspective. Atmospheric Chemistry and Physics, 13(14), 7053-7074. Zheng, J., Hu, M., Peng, J., Wu, Z., Kumar, P., Li, M., ... & Guo, S. (2016). Spatial distributions and chemical properties of PM2.5 based on 21 field campaigns at 17 sites in China. Chemosphere, 159, 480-487.) Therefore, the official report from the local government acquired based on long-term field survey and the relevant reference could support "The main source for NO3- is vehicle exhaust in Beijing".

Q14. L320-321. As "The main source for NO3- is vehicle exhaust" and the vehicles that cannot meet the Environmental Levels I and II was forbidden during orange alerts, why the concentration of NO3- was much higher during orange alerts in Mar, 2018 than that in March, 2013 without emission-reduction (Table 4)? Increased NO3- corresponded to deceased concentration of NH4+ during pollution episodes, so what's possible existing form of NO3- in PM2.5?

R : Thanks for pointing this out. We are sorry that the numbers in the previous manuscript were of some errors and we have corrected these numbers in the updated Table 5. Meanwhile, the OC and EC data you suggested were also added to Table 4.

Yes, as you suggested, the  $NO_3^-$  was actually in March, 2013(without emission-reduction measures) was much higher than that in March, 2018 (with "2+26" strategy). Thanks very much again for this correction.

Q15. L364-365. Please clarify what changes have been made to the air pollutants emission after "Coal to Gas".

R: Thanks so much for this comment. As we know, PM2.5 concentrations were highest in winter in Beijing, mainly due to the central heating required burning of coal materials. And this is the main reason for the high  $SO_2$  concentrations for wintertime  $PM_{2.5}$  component. After "Coal to Gas", a majority of coal ovens were replaced with equipment for gas burning, which led to less SO2 emissions and more oxynitride emission (http://www.sohu.com/a/208975915 801814) during central heating seasons. Based on the official assumption, the " coal to gas" project can lead to a 2 million-ton decrease in coal consumption in the Beijing-Tianjin-Hebei region.

Q16. L378-383. According to Fig.6, the local emission contributed 49.46% - 88.35% to PM2.5 during four pollution episode, indicating the local emission had a great effect on PM2.5 in Beijing. This is contradicting L92-93. And the different emission-reduction strategies did not lead to a clear pattern for the regional transport. So the PM2.5 reduction really was a result of "2+26" strategies or the meteorological condition? Whether the strict regulation on vehicle exhaust will be more effective than that of regional emission control under specific wind direction? The meteorological condition should be analyzed in detail for each pollution episode.

R: This is a very good point. As we stated in L92-93, although strict emission-reduction measures were conducted during two red alert periods, local  $PM_{2.5}$  concentrations remained high. We attributed this mainly to the large contribution of regional transport, which was not fully correct. Actually, in addition to different emission-reduction measures, meteorological factors and regional transport of airborne pollutants, the initial  $PM_{2.5}$  concentrations were crucial for the effects of emission-reduction measures, and the relative contribution of local emission and regional transport. So thanks a lot for pointing this out and we have corrected L92-93 accordingly in the revised manuscript.

This research found that the relative contribution of local emission and regional transport to PM2.5 concentrations in Beijing varied from 49.46%-88.35%, indicating the relative contribution of regional transport varied significantly. However, local emissions constantly made a large contribution to PM2.5 concentrations in Beijing. Some studies have proved that relative contributions of meteorological conditions and emission-reductions to PM2.5 concentrations in Beijing from 2013 to 2017 were around 20% and 80% (Chen et al., 2019). Similarly, according to a recent formal governmental report based on a series of studies (https://m.21jingji.com/article/20190311/herald/263828cd8f4cf3986ee1c39378c64881.html?fr om=groupmessage&isappinstalled=0), the relative contributions of meteorological conditions to PM2.5 concentrations were 10-15% during heavy pollution episodes. In addition, meteorological conditions during each pair of corresponding pollution episodes were similar. Therefore, PM2.5 reduction was mainly attributed to emission-reduction measures, including local and regional emission-reduction measures. In this case, since the relative contribution of vehicle bursts and local emissions were increasing notably in heavy pollution episodes (high PM2.5 concentrations), we suggested that strict regulations on vehicle exhaust should be effective ways for further reducing PM2.5 concentrations. As

explained above, the major meteorological influencing factor for  $PM_{2.5}$  concentrations (wind speed and relative humidity) in each corresponding episode were similar, so the  $PM_{2.5}$ concentration reduction were induced by different emission-reduction measures.

Q17. Overall, some of the conclusions on page 20 appear to be speculation with little data or discussion to support it, such as L494-495. Analysis and discussion on regional distribution of PM2.5 needs to be supplemented.

R: Thanks so much for this comment. A map of the distribution of  $PM_{2.5}$  concentrations during four pollution episodes and relevant discussion were added to the revised manuscript. As shown in Fig 6, the spatial distribution of PM2.5 concentrations in the "2+26" region may vary significantly during different pollution episodes. Therefore, the influence of regional long-term transport of PM2.5 concentrations on PM2.5 concentrations was controlled by the direction and intensity of PM2.5 transport and the comparison between PM2.5 concentrations in Beijing and upwind areas.

---

## Author Comment (AC2) · 14 Apr 2019

**To reviewer 2:**

**Thanks so much for your general and detailed comments. We have fully revised this manuscript according to these comments. We are more than willing to conduct further revisions if additional requirements are given.**

1. It is dangerous to evaluate the pollution control strategy by using only four pollution episodes. Too many parameters, especially the meteorological parameters, can influence the pollution level in one case, and would result in large uncertainties in the evaluation. A comparison based a long-period observation is needed. The current comparison between every two episodes, at least, not statistical significant.

**R:   This is a very good question. Thanks for pointing this out. Actually, the "2+26" strategy and regional air pollution alert with were contingent and implemented for severe pollution episodes. Therefore, the evaluation of short-term contingent local and regional emission-reduction measures were mainly conducted based on the analysis and simulation of PM$_{2.5}$ concentrations during short pollution episodes with different emission reduction measures. In this case, a large amount of studies (Jia et al., 2017; Cheng et al., 2017; Wang, et al., 2019; etc) were conducted simply based on one or two pollution episodes to evaluate the effects of different emission-reduction measures. On the other hand, as you pointed out, to evaluate long-term emission-reduction policies, instead of contingent emission-reduction measures, a long-term simulation should be conducted, which is another type of research based on other statistical methods ( e.g. Chen et al., 2019).**

**Chen, Z., Chen, D., Kwan, M., Chen, B., Cheng, N., Gao, B., Zhuang, Y., Li, R., and Xu, B.: The control of anthropogenic emissions contributed to 80 % of the decrease in PM2.5 concentrations in Beijing from 2013 to 2017, Atmos. Chem. Phys. Discuss., https://doi.org/10.5194/acp-2018-1112, 2019.**

**Cheng, N., Zhang, D., Li, Y., Xie, X., Chen, Z., M, F., Gao, B.B., He, B.: Spatio-temporal variations of PM2.5 concentrations and the evaluation of emission reduction measures during two red air pollution alerts in Beijing, Scientific Reports, 7(1), 8220,2017.**

**Wang Q, Liu S, Li N, et al. Impacts of short-term mitigation measures on PM 2.5 and radiative effects: a case study at a regional background site near Beijing, China[J]. Atmospheric Chemistry and Physics, 2019, 19(3): 1881-1899.**

**Jia J, Cheng S, Liu L, et al. An Integrated WRF-CAMx Modeling Approach for Impact Analysis of Implementing the Emergency PM2. 5 Control Measures during Red Alerts in Beijing in December 2015. Aerosol and Air Quality Research, 2017, 17: 2491-2508.**

2. In Fig. 2, it seemed to me the simulation result was too good. And the model can only underestimate PM2.5, but not overestimate, why? The author need to provide the comparison results of the chemical composition, but not only the mass concentration of PM2.5.

**R:Thanks for pointing this out. Actually, the $PM_{2.5}$ simulation result was satisfactory, but not too good compared with similar studies. Maybe the plot figure caused this confusion. According to your comment, we also added the simulation of meteorological factors and chemical compositions and presented a comprehensive accuracy assessment table in the revised manuscript. Thanks again for this point. The WRF-CAMx model generally underestimate $PM_{2.5}$ concentrations, not every day (For some days, the observed $PM_{2.5}$ concentrations can be lower than the simulated values). But for a heavily polluted episode, the averaged simulated $PM_{2.5}$ mass concentrations were generally lower than observed $PM_{2.5}$ concentrations, which was revealed by relevant studies. The possible reason for the underestimation of $PM_{2.5}$ concentrations using WRF-CAMx model might be attributed to this: the emission inventories for running this model, including industry and other categories, could not fully reflect the actual emission scenarios. Firstly, not all emission-sources can be included in the emission inventories. Secondly, the contingent emission-reduction measures during pollution episodes may not be fully implemented by all factories. Therefore, the actually emitted precursors were more than model-predicted and thus the WRF may underestimate $PM_{2.5}$ concentrations.**

Specific comments:

1. Remove "recent" in the title

**R: Corrected**

2. I would not recommend use 'Orange air pollution alert' in the title.

**R: Corrected.**

3. In Fig.3, this kind of direct comparison between two cases at different time did not make much sense.

**R:Actually, the "2+26" regional emission-reduction strategy for improving air quality in**

Beijing was recently proposed contingent policy and just implemented for twice. Therefore, to fully evaluate the effects of "2+26" strategy on $PM_{2.5}$ reduction, we selected two pollution episodes, one in March, 2013 with no emission-reduction measures and one in November, 2016 with local emission reduction measures to compare with the two pollution alerts with "2+26" emission-reduction measures, one in November, 2017 and one in March, 2018. Since the major meteorological conditions, initial $PM_{2.5}$ concentrations and the month between the pollution episodes in March, 2013 with no emission reduction measures and March, 2018 with "2+26" emission-reduction measures, and the pollution episodes in November 2016 with local emission-reduction measures and November 2017 with regional emission-reduction measures were generally similar. Therefore, comparing the corresponding pollution episodes were an effective approach for understanding the effects of local emission-reduction measures and regional emission-reduction measures on improving air quality in Beijing during pollution episodes. That is the reason we employed four pollution episodes to demonstrate the effects of "2+26" regional emission-reduction VS No emission-reduction, and "2+26" regional emission-reduction VS local emission-reduction.

4.In Fig. 4b, why there were such a high concentration of nitrite and chloride?

R: Thanks for pointing this out. Due to data recording errors, the $NO_3^-$ in the previous manuscript was wrongly used as the nitrite. We corrected this and added additional OC and EC to the revised manuscript. The updated figure was listed as follows. Thanks again for pointing this out and we are very sorry for this confusion.

[Figure]

**Fig 4. The variation of PM2.5 components in Beijing during four pollution episodes**

5. In Fig. 5, compared to the previous pollution episodes, the contribution of coal combustion in March 2018 episode decreased a lot, but the November 2017 case did not, why?

**R: As we know, PM$_{2.5}$ concentrations were highest in winter in Beijing, mainly due to the central heating (from November to March) required burning of coal materials. Since November, 2017, a large scale project "Coal to Gas" were implemented in the Beijing-Tianjin-Hebei region and a majority of coal ovens were replaced with equipment for gas burning in the "2+26" region, leading a notable decrease of the relative contribution of coal combustion to PM$_{2.5}$ concentrations. Based on the official assumption, the " coal to gas" project can lead to a 2 million-ton decrease in coal consumption in the Beijing-Tianjin-Hebei region.**